# Recommendations on Complementary Feeding as a Tool for Prevention of Non-Communicable Diseases (NCDs)—Paper Co-Drafted by the SIPPS, FIMP, SIDOHaD, and SINUPE Joint Working Group

**DOI:** 10.3390/nu14020257

**Published:** 2022-01-07

**Authors:** Margherita Caroli, Andrea Vania, Maria Carmen Verga, Giuseppe Di Mauro, Marcello Bergamini, Barbara Cuomo, Rosaria D’Anna, Giuseppe D’Antonio, Iride Dello Iacono, Angelica Dessì, Mattia Doria, Vassilios Fanos, Michele Fiore, Ruggiero Francavilla, Simonetta Genovesi, Marco Giussani, Antonella Gritti, Dario Iafusco, Lucia Leonardi, Vito Leonardo Miniello, Emanuele Miraglia Del Giudice, Filomena Palma, Francesco Pastore, Immacolata Scotese, Giovanni Simeone, Marco Squicciarini, Giovanna Tezza, Ersilia Troiano, Giuseppina Rosa Umano

**Affiliations:** 1Independent Researcher, Francavilla Fontana, 72021 Brindisi, Brindisi, Italy; margheritacaroli53@gmail.com; 2Independent Researcher, 00162 Rome, Rome, Italy; 3ASL Salerno, Vietri Sul Mare, 84019 Salerno, Salerno, Italy; vergasa@virgilio.it; 4ASL Caserta, Aversa, 81031 Caserta, Caserta, Italy; presidenza@sipps.it; 5AUSL Ferrara, 44121 Ferrara, Ferrara, Italy; marcelloberga54@gmail.com; 6Department of Pediatrics, Belcolle Hospital, 01010 Viterbo, Viterbo, Italy; cuomoba@gmail.com; 7Associazione Italiana Genitori (AGE), 00165 Rome, Rome, Italy; presidente@age.it; 8Independent Researcher, 84100 Salerno, Salerno, Italy; gdantonio32@gmail.com; 9Independent Researcher, 82100 Benevento, Benevento, Italy; iridello@gmail.com; 10Department of Surgical Sciences, University of Cagliari, 09100 Cagliari, Cagliari, Italy; angelicadessi@hotmail.it (A.D.); vafanos@tiscali.it (V.F.); 11AULSS 3 Serenissima, 30015 Chioggia, Venice, Italy; mattiadoria@fimp.pro; 12ASL3 Genovese, 16130 Genoa, Genoa, Italy; docmicky@libero.it; 13Interdisciplinary Department of Medicine, Aldo Moro University, 70100 Bari, Bari, Italy; rfrancavilla@gmail.com; 14IRCCS Auxologico, 20145 Milan, Milan, Italy; simonetta.genovesi@unimib.it (S.G.); abrjg@tin.it (M.G.); 15Dipartimento Scienze Formative, Psicologiche e Della Comunicazione, Università Suor Orsola Benincasa, 80135 Naples, Naples, Italy; antonellagritti.01@gmail.com; 16Department of Women, Children, and General and Specialist Surgery, University of Campania “Luigi Vanvitelli”, 80135 Naples, Naples, Italy; dario.iafusco@unicampania.it (D.I.); emanuele.miragliadelgiudice@unicampania.it (E.M.D.G.); giusi.umano@gmail.com (G.R.U.); 17Maternal Infantile and Urological Sciences Department, Sapienza University, 00161 Rome, Rome, Italy; lucialeonardi@yahoo.it; 18Policlinic Hospital Giovani XXIII, 70100 Bari, Bari, Italy; vito.miniello@libero.it; 19ASL Salerno, 84091 Battipaglia, Salerno, Italy; menapalma3@gmail.com; 20ASL Taranto, 74015 Martina Franca, Taranto, Italy; francesco.pastore60@tin.it; 21ASL Salerno, 84022 Campagna, Salerno, Italy; scotese.ped@libero.it; 22ASL Brindisi, 72023 Mesagne, Brindisi, Italy; giovanni.simeone@gmail.com; 23BLSD Training Activities of the Ministry of Health, 00144 Rome, Rome, Italy; msquicciarini@alice.it; 24F. Tappeiner Hospital, 39012 Merano, Bolzano, Italy; giovanna.tezza@gmail.com; 25Direzione Socio-Educativa, Municipio Roma III Montesacro, 00137 Rome, Rome, Italy; ersilia.troiano@gmail.com

**Keywords:** complementary feeding, recommendation, prevention, early nutrition, human milk, cow milk, food allergy, responsive feeding, BLW, BLISS

## Abstract

Adequate and balanced nutrition is essential to promote optimal child growth and a long and healthy life. After breastfeeding, the second step is the introduction of complementary feeding (CF), a process that typically covers the period from 6 to 24 months of age. This process is, however, still highly controversial, as it is heavily influenced by socio-cultural choices, as well as by the availability of specific local foods, by family traditions, and pediatrician beliefs. The Società Italiana di Pediatria Preventiva e Sociale (SIPPS) together with the Federazione Italiana Medici Pediatri (FIMP), the Società Italiana per lo Sviluppo e le Origine della Salute e delle Malattie (SIDOHaD), and the Società Italiana di Nutrizione Pediatrica (SINUPE) have developed evidence-based recommendations for CF, given the importance of nutrition in the first 1000 days of life in influencing even long-term health outcomes. This paper includes 38 recommendations, all of them strictly evidence-based and overall addressed to developed countries. The recommendations in question cover several topics such as the appropriate age for the introduction of CF, the most appropriate quantitative and qualitative modalities to be chosen, and the relationship between CF and the development of Non-Communicable Diseases (NCDs) later in life.

## 1. Introduction

Adequate and balanced nutrition is indispensable for the optimal growth of neonates, infants, and young children and is also a crucial prerequisite for a long and healthy life. Proper nutrition starts with breastfeeding, which occurs similarly in all countries of the world, and whose many positive effects, even in the long term, are still being investigated and increasingly confirmed. Complementary feeding (CF) begins in the second semester of life and is still heavily influenced by socio-cultural choices, availability of specific local foods, family traditions, and, last but not least, pediatricians’ beliefs. It is thus no surprise that CF is still highly controversial. CF is a process that begins when human milk (HM) or formula can no longer meet the energy and nutrient needs of an infant who is then gradually given new foods and textures. CF typically covers the period 6–24 months of age. Scientific interest in early childhood nutrition gained momentum from the recognition that during the first 1000 days of life, from fetal life onwards, nutrition influences much of a child’s health, right through to adulthood [1]. Most research efforts have focused on the long-term effects of breastfeeding. Indeed, breastfed infants have a lower infectious morbidity and mortality, and, later in life, show a higher intelligence when compared with non-breastfed infants. There is also growing evidence that breastfed infants are protected against being overweight and diabetes development later in life. Among the positive effects of breastfeeding, we cannot forget the protection against the development of breast cancer for mothers and probably also against the development of ovarian cancer [2]. On the contrary, few studies have looked at complementary foods, which become the core of an infant’s diet after a few months of life.

The mission of the Italian Society of Preventive and Social Pediatrics (Società Italiana di Pediatria Preventiva e Sociale—SIPPS) is to promote the psychophysical health of children, reduce social inequalities, and prevent the development of chronic degenerative diseases in adulthood. An essential step in achieving these ambitious goals is the development of evidence-based recommendations.

SIPPS has therefore promoted the drafting of complementary feeding-related recommendations, involving several Italian scientific societies with an interest and expertise in:i.Children’s primary care: Italian Pediatricians’ Federation (Federazione Italiana Medici Pediatri—FIMP);ii.Analysis of the origins of chronic-degenerative diseases: Italian Society for Developmental Origins of Health and Diseases (Società Italiana per lo Sviluppo e le Origini della Salute e delle Malattie—SIDOHaD);iii.Pediatric nutrition: Italian Society of Pediatric Nutrition (Società Italiana di Nutrizione Pediatrica—SINUPE).

These recommendations are intended to be a tool, based on rigorously evidence-based scientific analysis, to be used by pediatricians in their efforts to promote the health of today’s children as well as that of tomorrow’s adults and seniors.

## 2. Aim of the Paper

To formulate recommendations about the appropriate age and quantitative and qualitative modalities for the introduction of complementary foods into the diets of infants aged 6–24 months. The aim is to provide a practical and updated tool.

## 3. Methodology

### 3.1. Users of the Paper

This paper is specifically addressed to family pediatricians, gastroenterologists, nutritionists, and dieticians, but it is also recommended for use by parents, caregivers, and childcare educators.

### 3.2. Setting

The paper is mainly addressed to primary care.

### 3.3. Working Groups

The project at the basis of this paper was developed as follows: Drafting of the project, definition of the general aims and specific objectives of the paper, establishment of the schedule for each phase, establishment of the coordination process, selection of topics to be covered, selection of methods for consultation, evidence selection, collection and analysis, and finally establishment of methods for reaching a consensus within the panel.

The working groups that contributed to all stages of the project were as follows:**Group in charge of developing the paper**, which organized and supervised the different stages, from design to final drafting;**Multidisciplinary and multi-professional panel (MD-MP Panel)**, which worked out the key questions (KQs), discussed evidence, and formulated the recommendations. The panel was divided into groups tasked with writing the text of the paper, one for each chapter, each with its coordinator;**Methodology group** that formulated the structured clinical questions (Population, Intervention(s), Comparison, and Outcome—PICO Framework), developed the strategy and searched for evidence, critical analysis of the literature, and extraction and compilation of relevant data. It also contributed to the drafting of the recommendations according to the GRADE method (Grading of Recommendations, Assessment, Development, and Evaluation) [3], prepared the questionnaire for a vote on the recommendations according to the Delphi method [4,5], and analyzed the results;**Draft management group**, which acquired and checked the contributions of the groups tasked with writing the different chapters and prepared the final complete draft;**Group of external reviewers**, consisting of specialists in pediatric nutrition. The external reviewers did not take part in any of the development and drafting phases of the paper, nor did they take part in the votes to approve the recommendations. The individual reviewers made a blind review of the paper.

The methodologist group and the MD-MP panel held a series of regular meetings. Notes were kept on the dates of these meetings and all preliminary versions of the paper.

The MD-MP panel included pediatricians, specialists in childhood nutrition, family and hospital pediatricians, specialists/experts in endocrinology, auxology, allergology, emergency medicine, epidemiology and research methodology, dieticians, nutritional biologists, child neuropsychiatrists, communication experts, and parents’ representatives.

The members of the groups were appointed by the scientific societies of the various scientific fields or by relevant associations.

### 3.4. Formulation of Clinical Questions

Clinical questions and related outcomes were identified for each of the different chapters of the paper.

The methodologists group then developed each question using the PICO framework, by identifying Population, Intervention(s), Comparison, and Outcome.

The narrative questions and PICOs were shared and discussed with the rest of the panel using the GRADE method [3,6,7].

The Panel identified and classified the outcomes in terms of importance: Only those categorized as *critical* or *important* were taken into account in the literature review and, subsequently, in the formulation and grading of the recommendations.

### 3.5. Searching for Scientific Evidence

The search for primary studies was first carried out from the evidence syntheses, evidence-based guidelines (GLs), and systematic reviews (SRs) and was then completed to include studies published later than those included in the SRs and those considered to be of interest.

#### 3.5.1. General Inclusion Criteria

Time frame:For GLs: Last 5 years;For SRs: Last 10 years;For studies: From the date of completion of the bibliography of the SRs included or, should this date not be available, no time limit;Language of publication: No limit;Population: Healthy, full-term, normal weight at birth, breast-fed and/or formula-fed children, age 6–24 months, living in western industrialized countries;Type of publication: GLs, practice GLs, government publications, SRs, meta-analyses, randomized controlled trials (RCTs), multicenter studies, observational studies, cohort studies, longitudinal studies;Relevance to the clinical question;Methodological validity: Assessed according to the minimum criteria described in the section “Analysis of Scientific Evidence”.

The search strategy was discussed and agreed upon by the group in charge of methodology.

Search and evaluation of scientific evidence and data extraction were carried out at least in duplicate; in the event of discrepancies, the decision was made following joint discussion amongst the methodologists.

#### 3.5.2. Guidelines Search

Databanks of GLs: Italian National Guidelines System (Sistema Nazionale Linee Guida—SNLG), National Institute for Health and Clinical Excellence (NICE), Scottish Intercollegiate Guidelines Network (SIGN), Guidelines International Network (G-I-N), Canadian Medical Association (CMA) Infobase, Australian Clinical Practice Guidelines, and New Zealand (NZ) Guideline Group;PubMed http://www.ncbi.nlm.nih.gov/pubmed;EMBASE https://www.embase.com;UpToDate https://www-uptodate-com;Scientific societies: Italian Society of Pediatrics (SIP), SINUPE, SIPPS, Italian Society of Pediatric Gastroenterology Hepatology and Nutrition (SIGENP), Italian Society of Pediatric Endocrinology and Diabetology (SIEDP), European Society of Pediatric Gastroenterology Hepatology And Nutrition (ESPGHAN), and North-America Society of Pediatric Gastroenterology Hepatology And Nutrition (NASPGHAN).

#### 3.5.3. Systematic Reviews and Studies Search

Databases of SRs: Cochrane Library, CDSR—Cochrane Database of Systematic Reviews, DARE—Database of Abstract of Review of Effects In Cochrane Reviews, Other Reviews, Trials;PubMed http://www.ncbi.nlm.nih.gov/pubmed;EMBASE https://www.embase.com;Manual Search;Experts’ Bibliography.

#### 3.5.4. Keywords for Population, Intervention/Exposure Factor, Outcome, and Search Strings

They are given in the Appendix A of the paper for each of the questions to be answered.

### 3.6. Selection of Studies

An initial selection was made based on titles and abstracts, excluding studies published only as abstracts. The full texts of the studies considered potentially relevant or for which more information was needed to determine relevance were then obtained. The studies selected were then assessed in terms of their relevance to the selection criteria.

For each KQ, the selection process was recorded in PRISMA (Preferred Reporting Items for Systematic Reviews and Meta-Analyses) flow charts [8].

### 3.7. Analysis of Scientific Evidence

Evidence was analyzed and evaluated based on checklists and validated criteria.

The analysis of GLs was conducted using the validated AGREE II (Appraisal of Guidelines for Research and Evaluation II) tool [9].

For the evaluation of other documents, the criteria defined by the Italian National Guideline System (Sistema Nazionale Linee Guida SNLG) were used:Relevance of the topic;Date of publication <3 years (application of this criterion was assessed on a case-by-case basis);Multidisciplinary and multi-professional composition of the panel of experts;Clear and detailed description of the methodology adopted in line with the standards adopted by the National Centre for Clinical Excellence (CNEC) to assess the quality of scientific evidence [10].

The analysis of the SRs was conducted using the validated AMSTAR 2 (Assessment of Multiple Systematic Reviews) tool [11]. The minimum criterion of validity: An overall assessment of low methodological quality.

The quality of randomized studies was assessed using the validated Cochrane Collaboration tool “Assessment of Risk of Bias” [12].

For observational studies, the Newcastle Ottawa Scales were used for cohort, case-control, and cross-sectional studies [13]. The minimum criterion of validity: Absence of significant biases.

Biases and confounding factors specific to pediatric nutrition research were taken into account when assessing the quality of the studies [14].

### 3.8. Data Extraction and Management

At least two authors extracted data independently, using a pre-defined scheme based on the PRISMA [8] checklist criteria and checked for accuracy of the data in question.

All data were entered into Review Manager 5 (Review Manager 2014) [15] for RCTs and meta-analyses assessment.

### 3.9. Effect Size

Dichotomous variables—outcomes were calculated as Relative Risk (RR), for prospective studies, or Odds Ratio (OR), for retrospective studies, with confidence intervals (CI) of 95%.

Continuous variables—mean differences (MD) with 95% CI were calculated for continuous variables.

### 3.10. Missing Data

Loss to follow-up and compliance with any interventions were assessed in all the studies included.

The results obtained were analyzed, as far as possible, by intention-to-treat analysis and per protocol. Only available data were used, no other data were entered when data were missing.

### 3.11. Evaluation of Heterogeneity

The heterogeneity across the studies was also investigated.

Methodological heterogeneity was assessed by examining the risk of bias while clinical heterogeneity was assessed by examining similarities and differences across the studies in terms of types of participants, interventions, and outcomes. Effect size and direction were calculated and tau^2^, I^2^, and Chi^2^ statistical methods were used to quantify the level of statistical heterogeneity across the studies in each analysis.

Particular caution was used when interpreting the results with high levels of heterogeneity.

### 3.12. Data Synthesis

Meta-analyses were performed when it was possible to aggregate data.

In the presence of significant statistical heterogeneity, data were combined using the Random effect model. Mantel Haenszel method was used for dichotomous results and inverse variance for continuous results.

### 3.13. GRADE Method 

The overall assessment of evidence quality (high, moderate, low, and very low), the definition of the direction (in favor, against the intervention), and of the strength (strong, weak) of the recommendations were made using the GRADE method [3,6,7].

The main determinants of the strength of the recommendations were as follows:Balance between desirable and undesirable effects;Overall evidence quality for the outcomes under consideration;Values and preferences;Costs (allocation of resources).

### 3.14. Approval of Recommendations

The recommendations were voted on according to the Delphi method via a blind questionnaire. Five possible answers were provided: *Strongly agree*, *agree*, *neither agree nor disagree*, *disagree*, and *strongly disagree*.

There are no ambiguous criteria when it comes to the approval of recommendations.

In several documents of good methodological quality, recommendations are approved if the percentage of agreement is 70–75% or more (strongly agree, agree).

In the case of “neither agree nor disagree”, “disagree”, and “strongly disagree” answers, respondents were asked to explain the reason(s) for their answers.

In any case, all the comments given to explain the reason for the disagreement were carefully considered and recorded, both those related to the content of the recommendation and those related to its formal correctness and clarity of presentation.

### 3.15. Softwares

RevMan 5.4.115 software was used to assess the methodological quality of the RCTs, the meta-analyses, and the related figures [15].

The GRADEpro GDT software, developed by the GRADE Working Group, was used to assess the overall quality of the evidence collected and of the relevant tables [16].

### 3.16. GRADE-ADOLOPMENT

The GRADE-ADOLOPMENT approach is an evolution of the GRADE method that allows assessing whether it is possible to adapt to one’s context or to adopt recommendations of existing GLs, to answer specific clinical questions [4].

In this paper, GRADE ADOLOPMENT was used to assess the possibility of adapting or adopting the recommendations of specific GLs on food allergy prevention.

### 3.17. Updates

The paper will be updated 3 years from now or when new evidence is published that implies changes in the recommendations.

### 3.18. Implementation

The paper will be presented at congresses, continuing education courses, and within pediatric discussion forums and mailing lists. In particular, pediatricians and professionals involved in the entire “perinatal path” (birthing process included), and nutrition experts will be widely informed.

Special attention will be given to the dissemination to the general population through the media, and to kindergartens’ canteen staff.

### 3.19. Conflict of Interest (COI)

Each of the members of the working groups subscribed to the disclosure of any potential conflicts of interest (COI) in the preliminary stages of the project and at the end of the project. COIs (declared at the end of the article) were managed as follows:▪The members of the methodology group did not have COI;▪The authors with potential COIs were not involved in the systematic review of evidence. On the contrary, they participated in all the other phases of the drafting process each one of them according to their specific expertise;▪The methodology team and authors without COI checked the correctness and consistency of each part of the paper and especially of the recommendations; each author was asked to vote and to express and explain their disagreement anonymously;▪The results of vote counting and, in particular, the reasons for any disagreement were jointly discussed to produce the final version of the conclusions and recommendations hereunder.

## 4. Key Questions and Recommendations


**
*Key Question: Does an energy intake above the recommended levels for infants and young children aged 6–24 months lead to different short-term and long-term nutritional and metabolic outcomes compared with an intake in line with recommended levels?*
**

**
*Key Question: Does an intake of carbohydrates exceeding the recommended levels for infants and young children aged 6–24 months lead to different short-term and long-term nutritional and metabolic outcomes than an intake in line with recommended levels?*
**

**
*Key Question: Does a protein intake exceeding the recommended levels for infants and young children aged 6–24 months lead to different short-term and long-term nutritional and metabolic outcomes than an intake in line with recommended levels?*
**

**
*Key Question: Does a fat intake above the recommended levels for infants and young children aged 6–24 months lead to different short-term and long-term nutritional and metabolic outcomes than an intake in line with recommended levels?*
**

**
*Key Question: Can an excessive salt intake during CF lead to hypertension later in life?*
**


The evaluation of the effects of the amounts of each nutrient taken in with complementary foods on short-term (growth, body’s iron reserve) and long-term (risk of overweight/obesity, type 2 diabetes [T2D], hypertension) nutritional and metabolic outcomes presents several limitations concerning the design and way the studies were conducted.

The currently available studies are only observational, as no RCTs have been conducted on CF, but only on formula feeding.

Many studies have a poor methodological quality which is mainly ascribable to:High loss to follow-up, frequently >50%,Imprecise assessment of the exposure factor (amount of nutrient intake during the period of CF),Unreliable detection of outcomes (self-reported weight and length/height),Failure to assess important factors as potential confounders, in particular: Total energy intake after 2 years of age, percentage of a specific nutrient concerning total energy intake, and physical activity.

Due to these limitations and because the results obtained are inconsistent across the studies, no robust and documented conclusions can currently be drawn on the medium- and long-term outcomes of the different intake levels of the individual nutrients during CF.

Recommendations are, therefore, essentially formulated based on age-specific recommended intakes. Each individual recommendation, according to the GRADE framework, must be formulated in such a way as to contain essential information, so all the recommendations are formulated on the basis of a format that meets this requirement.

Thanks to improvements in the techniques for determining BMR and total energy expenditure, the current values of the energy requirements have been greatly reduced compared to those specified by WHO (World Health Organization) in 1985 [17], especially in the age group of 4 months–2 years, down to an almost 25% reduction at one year of age, as it was recommended in 2004. The values in question are considered as ARs and are therefore indicative of an average intake for the population.

The requirement for carbohydrates depends on the amount of protein and fat intake and changes considerably from 6 to 24 months [18]. The recommendations for carbohydrate intake in the first few months are based on the carbohydrate content of human milk (HM) and increases gradually up to 2 years of age with a concurrent reduction in fat intake as the child’s diet comes to be more similar to that of an adult [18]. With regard to carbohydrates in general, a special emphasis should be placed on added simple sugars. According to EFSA (European Food Safety Authority) [18]: “Although there is some evidence that high intakes (>20 E%) of sugars may increase serum triglyceride (TG) and cholesterol concentrations, and that >20 to 25 E% might adversely affect glucose and insulin response, the available data are not sufficient to set an upper limit for (added) sugar intake”. The intake of added simple sugars, mainly through consumption of sugar-sweetened beverages, does however correlate with increased energy intake, reduced nutritional quality of other foods, and development of NCDs, including obesity and dental caries [19]. Therefore, it is more prudent to keep the intake of simple added sugars as low as possible, especially at this sensitive age when metabolic patterns and eating habits are being set.

The recommended protein intakes as reported by EFSA [18] are based on data from the 2007 WHO/FAO (Food and Agriculture Organization)/UNU (United Nations University) publication [20]. The values referred to in the recommendations are expressed as PRI (M ± 2SD) and are, therefore, sufficient to cover the needs of 97% of the target population. While no institution/organization has yet identified any evidence-based higher level, it is generally accepted that “in adults an intake of twice the PRI is considered safe” [18]. However, in the first 24 months it seems that an excessive protein intake might promote the differentiation of preadipocytes into adipocytes by increasing their absolute number which, when combined with a slight surplus of dietary energy, would also promote excessive weight gain later in life [21]. These data have led to setting a maximum limit value of 14% energy intake from protein for this age group [22].

In the first six months of life, fat intakes meet approximately 50% of the energy requirements, in the second half of the first year their contribution to meet energy requirements decreases to around 40%, and then to around 30–35% at the end of the third year of life [23]. These high requirements result from the essential role of fats in the organic and functional development of the brain. There is no evidence that reducing fat intake in the first two years of life has any protective effect against the development of obesity or other NCDs later in life.

Diet-induced hyponatremia does not occur in healthy subjects because dietary sodium is more than enough to meet body requirements [24]. For the first six months of life, EFSA has established that the correct Na intake is that provided by HM or by HM substitutes, i.e., approximately 120 mg/day, while between 6 and 24 months of age an intake of 170 mg to a maximum of 370 mg/day is recommended [25]. A preference for savory taste develops around 4 months of age [26], which may, in the case of a family habit of excessive salt consumption and due to the phenomenon of tracking, also influence preference and consumption of excessive salt later in life [27]. Except for an old study on the first 6 months of life [28], no RCTs on salt intake in the first two years of life are available as safety data on excess salt intake are still not available. In adults, high salt intake is associated with increased hypertension and mortality, and an association between increased blood pressure and salt intake has been demonstrated in school-aged children, while a correlation between reduced dietary Na and lower blood pressure values has been demonstrated [29]. Moderate salt consumption is also recommended for the prevention of malignancies, in particular gastric cancer [30].


***Energy intake*—Recommendations**



**As no reliable data are available on the absence of short-, medium-, and long-term outcomes in healthy infants and young children with good weight and length gain, we recommend that daily energy intake remains in the range of the energy intake levels observed in the age-related healthy population groups reported by International Organizations/Societies, also considering the amount of physical exercise. (Expert opinion. Strong recommendation. Panel consensus 91%).**



***Carbohydrates*—Recommendations**


2.
**As no reliable data are available on the absence of short-, medium-, and long-term outcomes in healthy infants and young children with good weight and length gain, we recommend that daily carbohydrate intake remains within the range observed in the age-related healthy population groups reported by International Organizations/Societies. (Expert opinion. Strong recommendation. Panel consensus: 91%).**
3.
**We suggest that healthy infants and young children with good weight and length gain do not exceed the requirements of carbohydrates (especially monosaccharides and disaccharides) with complementary foods to prevent medium- and long-term outcomes, including overweight and obesity later in life. (Low Quality of evidence. Weak recommendation. Panel consensus: 100%).**



***Protein*—Recommendations**


4.
**We recommend that in healthy infants and young children with good weight and length gain, daily protein intake remains in the range observed in healthy population groups reported by International Organizations/Societies. (Expert opinion. Strong recommendation. Panel consensus: 91%).**
5.
**We do not suggest that the protein intake during the period of CF (6–24 months) exceed 14% of total energy levels observed in age-related healthy population groups to prevent short-, medium-, and long-term outcomes including being overweight and obesity later in life. (Low quality of evidence. Weak recommendation [against]. Panel consensus: 100%).**



***Fats*—Recommendations**


6.
**We recommend that in healthy infants and young children with good weight and length gain, the daily intake of lipids remains in the range observed in healthy population groups reported by international Organizations/Societies. (Expert opinion. Strong recommendation. Panel consensus: 100%).**
7.
**We do not recommend that during the period of CF healthy infants and young children with good length and weight gain take fewer lipids than the recommended intake for their age to prevent medium- and long-term outcomes including being overweight and obesity later in life. (Moderate quality of evidence. Strong recommendation [against]. Panel consensus: 100%).**



***Salt*—Recommendations**


8.
**Based on currently available evidence on the correlation between salt intake and risk of developing hypertension in childhood and adulthood, and in the absence of reliable data on the intake of salt added to foods during the CF period that would exceed the infant’s requirements, we recommended that no salt be added to foods for at least the first year of life, and preferably also in early childhood, as long as the amount of salt that is naturally contained in foods corresponds to the age-related recommended levels. (Low quality of evidence. Strong recommendation [against]. Panel consensus: 100%).**


Additional data for recommendations 1–8 are in Appendix A (link at the end of the paper).


**
*Key Question: Does starting CF between 4 and 6 months of age result in different short-term and long-term nutritional and metabolic outcomes compared to exclusive breastfeeding for the first 6 months of life?*
**

**
*Key Question: Does starting CF between 4 and 6 months of age result in different short-term and long-term nutritional and metabolic outcomes compared to exclusive formula feeding or mixed feeding (human milk + formula) for the first 6 months of life?*
**


Since 2005, WHO has recommended that both breastfed and formula-fed infants should start CF at six months [31]. By contrast, while recommending exclusive breastfeeding up to 6 months of age, both ESPGHAN and AAP (American Academy of Pediatrics) state that it is possible to start CF between 17 and 26 weeks of age [32,33], thus not helping to define the appropriate age for introducing CF. EFSA has aligned its scientific opinion on the duration of breastfeeding with WHO recommendations, but has not issued recommendations on the age at which CF should be introduced [34]. The ability to start CF does not only depend on the maturation of the intestinal wall and of the functional development of the kidney, but also and above all on the achievement of some important neurodevelopmental milestones including the ability to (i) sit without support, (ii) express hunger and satiety signals, and (iii) reach for spoon and food, all abilities that 97% of infants achieve around six months of age. By the start of CF, all of these milestones must have been reached. On the other hand, the total nutrient adequacy of both HM and formulas in the first six months has been amply demonstrated by WHO and EFSA [34,35]. Indeed, based on the analysis of the best available scientific evidence, infants fed HM or formula, irrespective of whether they had started CF between 4 and 6 months or at the end of six months of age, have not shown differences in both weight and length gain and iron status at the age of 12 months, and no differences have also been shown in terms of the risk of developing overweight/obesity, T2D, and hypertension in the long term.

Last, but not least, for breast-fed infants introducing other foods before 6 months of age may lead to the risk of a lower intake of HM and, consequently, a reduced intake of functional substances that play a crucial role and support the optimal growth and development of the infant’s brain, immune system, etc.

Finally, as also reported in the EFSA SR [34], the mere fact that solid foods can be introduced before 6 months of age does not necessarily imply that introducing such foods is imperative or desirable.


**Recommendations**


9.
**We recommend that in healthy breast-fed infants with good length and weight gain, CF does not be introduced before 6 months of age, taking into account the specific non-nutritional benefits of HM (maternal antibodies, stem cells, growth factors, microbiota). (Moderate quality of evidence. Strong recommendation [against]. Panel consensus: 94.4%).**
10.
**Should the mother of healthy breast-fed infants with good length and weight gain, be unable to continue exclusive breastfeeding between 4 and 6 months of age of the infant due to specific needs shared and discussed with her baby’s pediatrician, we suggest that options for supplementation be considered, with the formula being preferred over complementary foods (Expert opinion. Optional recommendation. Panel consensus: 71.5%).**
11.
**We suggest that in healthy formula-fed infants with good length and weight gain, CF should not be introduced before 6 months of age. (Moderate quality of evidence. Weak recommendation [against]. Panel consensus: 76.1%).**
12.
**Without prejudice to the recommendation on the introduction of complementary foods at 6 months, no other age or time frame is recommended, e.g., before 4 months or after 6 months. (Strong recommendation [against]. Panel consensus 95.2%).**
13.
**We suggest that in healthy breastfed or formula-fed infants the age when infants start receiving CF (specifically for the two options: 4–6 or 6 months) does not need to be “used” as a preventive measure to control NCDs: Overweight/obesity, T2D, and hypertension. Weak recommendation [against]. Panel consensus: 90.4%).**


Additional data for recommendations 9–13 are in Appendix A (link at the end of the paper).


**
*Key Question: Does receiving cow milk (CM) before 12 months of age, compared to formula feeding, result in different short-term and long-term nutritional and metabolic outcomes?*
**

**
*Key Question: Does receiving unmodified CM after 12 months of age, compared to Young Child Formula (YCF), result in short-term and long-term adverse metabolic effects?*
**


The composition of CM differs a lot from that of HM and does not meet all the nutritional needs of the infant. It cannot, therefore, be considered a ‘complete’ food for the human species, especially in the first six months of life when it is the only food for babies.

In the absence of HM, if the infant needs to receive CM-derived formula, its macronutrient composition should be as similar as possible to that of HM to provide similar health effects.

After one year of age, a child’s diet should include all foods, with a reduction in milk consumption. However, CM can only be considered appropriate when combined with other foods to compensate for both deficit and excess of micro- and macro-nutrients.

Currently, the available evidence is mainly based on observational studies, with very few RCTs focused on the comparison between the effects of CM and the effects of formulas in children both under and over 12 months of age.

The results regarding the auxological parameters are mixed, but most studies report no significant differences for CM intakes of less than 600 mL/day.

All the studies on iron-deficiency anemia (IDA) in infants and small children agree in recommending not to use unmodified CM in the first year of life. Only one observational study shows a possible dose-response gradient for an alteration in the body’s iron reserves for intakes of unmodified CM exceeding 500 mL/day before 18 months of age.

The results obtained on the potential association between the intake of unmodified CM in the first year of life and the development of type 1 diabetes [T1D] are not consistent. However, these studies have poor methodological quality and therefore no conclusions can be drawn from them.


**Recommendations**


14.
**We recommend that for infants up to 12 months of age who need to supplement in part or completely replace HM, the unmodified CM do not be given as an alternative to formulas (high quality of evidence for the risk of IDA, low for the risk of developing T1D, low for auxological parameters). (Strong recommendation [against]. Panel consensus: 100%).**
15.
**We suggest that children aged 12–24 months who need to supplement in part or completely replace HM, and who receive different nutrients at recommended levels, can be fed unmodified CM. (Moderate quality of evidence. Weak recommendation. Panel consensus 85%).**
16.
**For children aged 12–24 months who need to supplement in part or completely replace HM and who are still on the main milk diet, we suggest the use of a formula as an alternative to unmodified CM (expert opinion. Weak recommendation. Panel consensus 100%), which is useful for both IDA prevention (weak recommendation. Panel consensus 100%) and for limiting protein intake (expert opinion. Weak recommendation. Panel consensus 100%).**
17.
**We recommend that the amount of CM taken by children aged 12–24 months who live in developed countries should be less than 500 mL/day. (Quality of evidence moderate. Strong recommendation. Panel consensus 75%).**
18.
**We recommend that children aged 12–24 months who consume CM, especially in amounts exceeding 500 mL/day, should undergo careful nutritional assessment. (Expert opinion. Strong recommendation. Panel consensus 100%).**


Additional data for recommendations 14–18 can be found in Appendix A (link at the end of the paper).


**
*Key Question: Can the Baby Led Weaning (BLW)/Baby-Led Introduction to Solids (BLISS) method during CF influence, either positively or negatively, the stature-ponderal growth process later in life?*
**


The spread of CF models based above all on the baby-caregiver relational aspects with declared positive effects also in terms of long-term metabolic outcomes calls for a comprehensive evaluation of these models.

The models under consideration are: BLW, based only on child-food relational aspects since parents are not offered nutritional information; BLISS: In addition to relational aspects this method takes into account some nutritional aspects (it is recommended that each meal includes a protein-rich food, an energy-rich food, and a fiber-rich food); and Responsive Complementary Feeding (RCF), also based on relational aspects, but with an emphasis on nutrition education to families.

The outcome indicators deemed to be of high importance were:▪General growth parameters;▪Risk of NCDs (overweight/obesity, diabetes, and hypertension);▪Risk of choking;▪Risk of dental caries.

The most robust evidence to answer the clinical question comes from two controlled studies in which the BLISS model was used. The results of these studies are somewhat mixed. What is more, the poor methodological quality of the two observational studies raises uncertainty about the results obtained, which appear to support the use of BLW to reach a healthy weight in the post-CF period.

In addition, it should be noted that in the studies in question, the risk of nutritional deficits was inadequately assessed via simple questionnaire surveys on micro-macronutrient intakes, food preferences, and food variety during the first 2 years of life, thus making the results even more uncertain.


**Recommendations**


19.
**We suggest not to use the BLW method to improve children’s growth processes given the lack of conclusive evidence of its effectiveness and the potential risks of malnutrition (very low quality of evidence. Weak recommendation [against]. Panel consensus 88.9%).**
20.
**We suggest not to use the BLISS method to improve children’s growth processes given the lack of conclusive evidence of its effectiveness (low quality of evidence. Weak recommendation [against]. Panel consensus 88.9%).**



**
*Key Question: Can the use of the BLW/BLISS method during CF influence, either positively or negatively, the development of overweight/obesity later in life?*
**


Currently, the available evidence is provided from two RCTs and two observational studies, all with poor methodological quality and conflicting results. Only one study indicates the prevention effectiveness of this method, but this needs to be confirmed by further studies or by aggregate data meta-analysis.


**Recommendations**


21.
**We suggest not to use the BLW and BLISS for prevention of pediatric obesity (low quality of evidence. Weak recommendation [against]. Panel consensus 100%).**



**
*Key Question: Can responsive feeding (RF) during the CF period influence, either positively or negatively, the physical growth process later in life?*
**

**
*Key Question: Can non-RF during the CF period influence, either positively or negatively, the physical growth process later in life?*
**


The literature studies on the potential influence of RF and non-RF practices during the CF period on children’s growth later in life are affected by several important biases, in particular by performance biases and by poorly monitored or fully unmonitored compliance. In addition, the outcome indicators used varied widely between studies, not only in terms of the type of measurement, but also in terms of the time of measurement. The overall quality of the evidence is therefore low.


**Recommendations**


22.
**Based on currently available evidence, we suggest that RF practices be promoted from the earliest months of a child’s life and then encouraged and reinforced during the period of CF, as this latter is likely to result in adequate weight gain during the first two years of life. (Low quality of evidence. Weak recommendation. Panel consensus: 100%).**
23.
**Concerning some CF practices characterized by non-responsive caregiver behaviors (restrictive non-responsive styles or forcing or pressuring or controlling/monitoring; restrictive, indulgent, rewarding styles; and lack of active involvement or true disinterest), based on currently available evidence it is not possible to give indications about their impact on the growth processes during the first years of life. (Very low quality of evidence. Panel consensus: 88.9%).**



**
*Key Question: Does RF influence the development of overweight and obesity later in life?*
**

**
*Key Question: Does non-RF influence the development of overweight and obesity later in life?*
**


From a general point of view, it is very important to point out that: (i) The interventions implemented in randomized trials have multiple components, start before six months of age and are not time-specific; (ii) in the control groups feeding practices similar to those in the intervention arm may occur quite unintentionally; and (iii) the general compliance and perseverance of the caregivers in implementing the educational instructions received were not carefully monitored in all studies.


**Recommendations**


24.
**We suggest that the RF practices be promoted from the first months of a child’s life and then encouraged and reinforced during the CF, as they are likely to contribute over time to reach an adequate weight during the first 2–3 years of life (moderate quality of evidence. Weak recommendation. Panel consensus: 100%).**
25.
**For CF practices characterized by non-responsive caregivers’ behaviors and therefore by relational gaps (restrictive non-responsive styles or forcing or pressuring or controlling/monitoring styles; restrictive, indulgent, rewarding styles; and lack of active involvement or true disinterest), based on currently available evidence it is not possible to give indications about their impact on potential future alterations of the nutritional status of the child, either in terms of over-nutrition or in terms of under-nutrition. (Very low quality of evidence. Panel consensus: 100%).**



**
*Key Question: Do different CF models (styles) result in a different risk of choking?*
**


a.
**
*BLW*
**


The studies on the relation between choking and the kind of CF are of various types and have low methodological quality. No difference in risk has been demonstrated across the different types of CF.


**Recommendations**


26.
**Currently available evidence suggests that BLW and BLISS practices do not lead to an increased risk of choking during meals. We suggest that no specific type of CF should be encouraged or avoided exclusively to reduce the risk of choking (moderate quality of evidence. Weak recommendation [against]. Panel consensus: 77.7%).**


b.
**
*Responsive Complementary Feeding (RCF)*
**


There is essentially no evidence for a cause-effect relationship between RCF or non-RCF and choking events.


**Recommendations**


27.
**Based on the evidence gathered, it is not possible to define whether different styles of CF, RCF, or non RCF, lead to a higher or lower risk of choking during meals. We suggest that a specific CF style should not be encouraged or avoided for the sole purpose of reducing the risk of choking (very low quality of evidence. Weak recommendation [against]. Panel consensus: 100%).**
28.
**Regardless of the style of CF adopted, we recommend that the infant should always be closely supervised during meals (expert opinion. Strong recommendation. Panel consensus: 100%).**



**
*Key Question: Does RCF influence the development of T2D later in life?*
**

**
*Key Question: Can traditional CF influence the development of T2D later in life?*
**


The systematic literature search did not find any published evidence on the influence of either RF or non-RF styles during CF on the outcome indicator for T2D.


**Recommendations**


29.
**Considering the current absence of relevant evidence, it is not possible to formulate recommendations or suggest alternatives when it comes to RCF and non-RCF interventions to prevent the development of T2D later in life (expert Opinion. Panel consensus 88.9%).**



**
*Key Question: Can RCF influence the development of hypertension later in life?*
**


The systematic literature search did not find any published evidence on the influence of either RF or non-RF styles during CF on the outcome indicator for hypertension.


**Recommendations**


30.
**Considering the current absence of relevant evidence, it is not possible to formulate recommendations or suggest alternatives when it comes to RCF and non-RCF interventions to prevent the development of hypertension later in life (expert opinion. Panel consensus: 100%).**



**
*Key Question: Can RCF influence the development of dental caries later in life?*
**

**
*Key Question: Can traditional CF influence the development of dental caries later in life?*
**


None of the four papers found via scientific societies’ websites and GLs databases search deals with the relationship between the development of dental caries and behaviors/relational styles adopted during the CF period. Even searching for primary clinical studies in pediatric age, without time limits, yielded negative results as no study was found to include an analysis of the influence of RCF or non-RCF on the development of dental caries. Thus, it is not possible to answer the question as there is no evidence to shed light on the possible role of RCF or non-RCF in the development of dental caries later in life.


**Recommendations**


31.
**Considering the current absence of relevant evidence, it is not possible to formulate recommendations or suggest potential options in terms of RCF and non-RCF interventions to prevent the development of dental caries later in life (expert Opinion. Panel consensus: 100%).**


Additional data for recommendations 19–31 are in Appendix A (link at the end of the paper).


**
*Key Question: Can the timing of gluten introduction affect the development of celiac disease (CD)?*
**


Currently, available evidence comes from RCTs and observational cohort and case-control studies as well as from SRs and meta-analyses. The results of these studies are generally consistent and show that the timing of gluten introduction has no preventive or risk-increasing effect on the development of CD.


**Recommendations**


32.
**We recommend that the timing of introducing gluten be neither brought forward nor delayed to prevent the onset of CD (high quality of evidence. Strong recommendation [against]. Panel consensus: 100%).**
33.
**We recommend introducing gluten at the beginning of the CF period together with other foods (high quality of evidence. Strong recommendation. Panel consensus: 100%).**



**
*Key Question: Is the development of CD influenced by the CF/type of milk feeding relationship?*
**

**Recommendations**


34.
**We suggest that BF, for which a strong recommendation exists, should not be used as a preventive measure to counter the development of CD in infants at risk (low quality of evidence for the duration of BF. Moderate quality of evidence for BF vs. no BF and BF at the time of gluten introduction. Weak recommendation [against]. Panel consensus: 100%).**


Additional data for recommendations 32–34 are in Appendix A (link at the end of the paper).


**
*Key Question: Can the timing of the introduction of potentially allergenic foods influence the development of a food allergy?*
**


The currently available scientific evidence comes from RCTs and SRs of mixed methodological quality, which ranges from “high-to-moderate” for some outcomes to “low-to-very low” for other ones.

As for the introduction of potentially allergenic foods, generally speaking, the panel of this paper updated the previous recommendations based on the SR by De Silva et al., since the 2021 update of the European Academy of Allergology and Clinical Immunology (EAACI) GLs did not include the relevant question anymore.

The only study that deals with this topic, which is of very low quality, does not justify either delaying or encouraging exposure to allergenic substances once CF has begun, regardless of the atopic risk.

As to the introduction of chicken egg and peanuts into infant’s diet, the panel of this paper decided to modify recommendation 3.1.2 (introduction of raw/cooked egg) referring it only to infants at risk of egg’s proteins allergy and to also adapt recommendation 3.1.3. of the EAACI 2021 GLs (introduction of peanuts in countries with a high prevalence of peanut allergy) to our population, despite the low prevalence of peanut allergy in our country.

The amendment of recommendation 3.1.2 of the EAACI GL and the formulation of recommendation 37 of this document have been much discussed, considering that:The study on the cooked egg is unique, so the results cannot be considered conclusive;The population consists of infants with atopic dermatitis, so the results cannot be automatically transferred to the general population;The study evaluates not only the effectiveness of the cooked egg but also administration with very low and increasing doses, doses that do not correspond to those administered in daily practice and is thus difficult to measure without adequate tools (difficult applicability);At the end of the intervention, only the tolerance to the cooked egg and not to the raw egg was tested (therefore, we can only talk about the prevention of allergy to the cooked egg and not to the egg in general).

Although the EAACI 2021 GLs state that “In these countries, peanuts should be included in the diet according to normal dietary habits and local recommendations”, considering that:Peanut proteins are contained in many foods commonly consumed in early childhood (snacks, creams) and that;Peanut products are also readily available for consumption by very young children (creams), in high-risk children, even those living in countries with a low prevalence of peanut allergy, small amounts of peanut-containing foods may be recommended for use even before 11 months of age.
**Recommendations**
35.**In healthy breastfed or formula-fed infants, we recommend the introduction of potentially allergenic foods at 6 months of age, irrespective of the type of milk and the atopic risk, without postponing or bringing exposure forward to reduce the risk of food allergy. (Low quality of evidence for cooked egg, moderate for peanuts, very low for other allergenic foods. Strong recommendation. Panel consensus: 77.8%).**36.**We recommend that potentially allergenic foods be introduced with the same modalities to both infants at risk of allergy and infants at no risk of allergy (strong positive recommendation. Quality of evidence low for cooked egg, moderate for peanut, very low for other allergenic foods. Panel consensus 88.9%).**37.**Only in children with severe atopic dermatitis, at risk of allergic disease, we suggest the possibility of introducing well-cooked chicken egg, but not raw or pasteurized uncooked egg, as part of the complementary diet, to reduce the risk of adverse reactions. Any specific schemes or methods of administration aimed at the prevention of egg allergy should be indicated by the allergist pediatrician (low quality of evidence for cooked egg, very low for raw or pasteurized egg. Weak recommendation. Panel consensus: 87.5%).**38.**In children at risk of allergic disease with severe atopic dermatitis or egg allergy, even those living in countries with a low prevalence of peanut allergy, the introduction of peanuts into the diet may be suggested no later than 11 months of age to reduce the risk of allergy to this food (moderate quality of evidence. Weak recommendation. Panel consensus: 88.9%).**

Additional data for recommendations 35–38 are in Appendix A (link at the end of the paper).

## 5. Conclusions

Although all recommendations were formulated on a solid scientific evidence basis, evidence quality turned out to be high in only three very specific and limited cases: (1) The strong recommendation on CM intake in the first year of life due to the risk of developing iron-deficiency anemia; (2) the strong recommendation on the early or delayed introduction of gluten to prevent the onset of CD; and (3) the strong recommendation on the introduction of gluten at the beginning of CF together with other foods.

Several conclusions, which are considered to be well-established, are indeed based on low-quality evidence, e.g., excessive intake of simple sugars or proteins as a risk factor for the development of obesity, or excessive salt intake as a risk factor for the development of hypertension, the use of BLISS to promote growth, the use of BLW or BLISS used to prevent obesity later in life, and the RF style as a tool to promote healthy growth and particularly weight gain in the first three years of life.

Finally, other interventions that are generally perceived as being effective and successful are based on very low-quality evidence: The use of BLW as a means to improve child growth, the role of non-RF styles on child growth or impaired nutritional status, the use of a specific CF relational style to reduce the risk of choking, and the duration of BF as a protection against the development of CD.

Therefore, there are important fields of research that should be more thoroughly explored in the nearest future:Role of individual nutrients in the development of NCDs later in life;Age or time window when a specific nutrient may act as a trigger for a programming process;Importance of the mechanism for tracking nutrients including salt and sugar;Real role of new CF styles (BLW, BLISS) in improving children’s growth, preventing obesity, and improving family eating styles;Real impact of responsive and non-responsive feeding styles on children’s growth and nutritional status in the first years of life;Risk of choking associated with different CF styles;Role of responsive vs non-responsive feeding styles in the development/prevention of NCDs.

In conclusion, pediatricians have to make the most of the protective potential of CF for children’s health and to this end, it is their responsibility to be aware of and share with families all currently available evidence and to do more research into the many fields that still face a wide range of uncertainties.

## Data Availability

Not applicable.

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
