# Peer review of "Recommendations on Complementary Feeding as a Tool for Prevention of Non-Communicable Diseases (NCDs)—Paper Co-Drafted by the SIPPS, FIMP, SIDOHaD, and SINUPE Joint Working Group"

_nutrients, 2022, doi:10.3390/nu14020257_

Round 1
Reviewer 1 Report
The manuscript "Recommendations on complementary feeding as a tool for prevention of non-communicable diseases (NCDs) – Paper co- 3 drafted by the SIPPS, FIMP, SIDOHaD, SINUPE Joint Working 4 Group 5" from Caroli et al, deals with the significant topic of complementary feeding in infancy.
Some suggestions for improvements
Lines 50-52 please consider rephrasing this part to become more formal
Lines 73-75 please be more specific on the health/overall benefits of breastfeeding in early life
Line 76 I believe it would benefit the text of you add the whole text in Italian as well for “Society of Preventive and Social Pediatrics (SIPPS)”
Lines 345-393 This part is highly significant and should be strengthened and become more specific that repetitive. For example please include here in short the current recommendations
Lines 400-415 the decision on the exact introduction time results from proper growth and overall developmental status of the child (including neurodevelopment/stature ponderal growth). Please rephrase to clarify your message
Line 491 please include the abbreviations’ explanation here
Lin 707 please rephrase the sentence-it is unclear
Overall the document must be revised for increasing the fluency and become more specific in the respective outcomes presented
Author Response
Please, see the attached file.

Reviewer 2 Report
The manuscript is very well-written and I have no real concerns at all with it.
So for me it is OK to accept .
Author Response
Dear reviewer, thanks a lot for your appreciation of our efforts, it means a lot to us. Best regards and best wishes.
Reviewer 3 Report
Scientific article with 38 recommendations on complementary feeding. An evidence-based recommendation methodology has been followed.
Publication is endorsed by several scientific societies.
Questions about complementary feeding have important scientific doubts and, historically, they have been elucidated following local cultural aspects, as the authors point out. There is little scientific evidence and for this reason, the conclusions are not obviously categorical.
The methodology is well described and the results have been described in an orderly fashion.
There are small errors when explaining the acronyms used. Some are defined correctly the first time they appear in the text, but others are not (GLs and SRs on line 150; Type 2 diabetes on line 327; YCF on line 443 ... it would be convenient to review all).
Although most of the recommendations do not have sufficient force of scientific evidence, I believe that it is an important and necessary work.
